# The Sugar Metabolic Model of *Aspergillus niger* Can Only Be Reliably Transferred to Fungi of Its Phylum

**DOI:** 10.3390/jof8121315

**Published:** 2022-12-17

**Authors:** Jiajia Li, Tania Chroumpi, Sandra Garrigues, Roland S. Kun, Jiali Meng, Sonia Salazar-Cerezo, Maria Victoria Aguilar-Pontes, Yu Zhang, Sravanthi Tejomurthula, Anna Lipzen, Vivian Ng, Chaevien S. Clendinen, Nikola Tolić, Igor V. Grigoriev, Adrian Tsang, Miia R. Mäkelä, Berend Snel, Mao Peng, Ronald P. de Vries

**Affiliations:** 1Fungal Physiology, Westerdijk Fungal Biodiversity Institute & Fungal Molecular Physiology, Utrecht University, Uppsalalaan 8, 3584 CT Utrecht, The Netherlands; 2Department of Biology, Concordia University, 7141 Sherbrooke Street West, Montreal, QC H4B 1R6, Canada; 3USA Department of Energy Joint Genome Institute, Lawrence Berkeley National Laboratory, 1 Cyclotron Rd, Berkeley, CA 94720, USA; 4Environmental Molecular Sciences Laboratory, Pacific Northwest National Laboratory, Richland, WA 99354, USA; 5Department of Plant and Microbial Biology, University of California Berkeley, Berkeley, CA 94598, USA; 6Department of Microbiology, University of Helsinki, Viikinkaari 9, 00014 Helsinki, Finland; 7Theoretical Biology and Bioinformatics, Utrecht University, Padualaan 8, 3584 CH Utrecht, The Netherlands

**Keywords:** sugar catabolism, orthology-based approach, metabolism network, transcriptome analysis

## Abstract

Fungi play a critical role in the global carbon cycle by degrading plant polysaccharides to small sugars and metabolizing them as carbon and energy sources. We mapped the well-established sugar metabolic network of *Aspergillus niger* to five taxonomically distant species (*Aspergillus nidulans*, *Penicillium subrubescens*, *Trichoderma reesei*, *Phanerochaete chrysosporium* and *Dichomitus squalens*) using an orthology-based approach. The diversity of sugar metabolism correlates well with the taxonomic distance of the fungi. The pathways are highly conserved between the three studied Eurotiomycetes (*A. niger*, *A. nidulans*, *P. subrubescens*). A higher level of diversity was observed between the *T. reesei* and *A. niger*, and even more so for the two Basidiomycetes. These results were confirmed by integrative analysis of transcriptome, proteome and metabolome, as well as growth profiles of the fungi growing on the corresponding sugars. In conclusion, the establishment of sugar pathway models in different fungi revealed the diversity of fungal sugar conversion and provided a valuable resource for the community, which would facilitate rational metabolic engineering of these fungi as microbial cell factories.

## 1. Introduction

Plant biomass is one of the most abundant renewable resources on earth, and it serves as a primary raw material in a variety of industries [1]. In nature, fungi can effectively release monosaccharides from plant biomass polysaccharides [2,3,4] and further metabolize them for energy and growth through a variety of sugar catabolic pathways [5]. D-glucose is by far the most abundant monosaccharide in plant biomass, and it is the major component of cellulose, starch and some hemicelluloses. D-fructose is found as a free monosaccharide in many fruits and plants, as well as being a major component of the disaccharide sucrose and polymer inulin [6]. Typically, D-glucose and D-fructose are easily taken up from the environment and catabolized by fungi, making them preferential monomeric carbon sources. D-fructose enters the cell and is phosphorylated to fructose 6-phosphate before entering glycolysis. D-glucose is phosphorylated to glucose 6-phosphate by glucokinase [7] and hexokinase [8], which can then be converted into fructose 6-phosphate. The pentoses D-xylose and L-arabinose are converted through the pentose catabolic pathway (PCP) to D-xylulose-5-phosphate, which enters the pentose phosphate pathway (PPP) [9]. Other sugars, like D-galactose, D-mannose, D-galacturonic acid, L-rhamnose and D-gluconic acid, are converted through sugar-specific metabolic pathways [5,10] and eventually all enter glycolysis. The final product of glycolysis is further metabolized through the tricarboxylic acid (TCA) and the glyoxylate cycles. The TCA cycle is present in almost all aerobic living organisms, including most fungi [11,12]. The glyoxylate cycle is essentially a truncated version of the TCA, which metabolizes acetyl-CoA without the loss of CO_2_ [13]. The inputs to both cycles are acetyl-CoA and oxaloacetate, which originate from the pyruvate of glycolysis. In short, fungal central carbon catabolism is a complex network that involves many pathways.

A systematical investigation of the components and diversity of sugar metabolic networks of different species is crucial for our understanding of the role of fungi in their natural environment as well as many fungal industrial applications. In a previous study of *A. niger* NRRL 3 [10], a model for the sugar metabolic pathways of this species has been established, which consists of 155 genes involved in 11 sugar metabolic pathways (glycolysis, PCP, PPP, D-galactose metabolism, D-mannose metabolism, D-gluconate metabolism, D-galacturonic acid catabolism, L-rhamnose metabolism, glycerol metabolism, TCA and glyoxylate cycles). Meanwhile, recent experimental studies have refined this predicted sugar metabolic network and identified new enzymes involved in L-rhamnose catabolism [14] and PCP [9].

In this study, we aimed to transfer the sugar metabolic model of *A. niger* to five taxonomically diverse fungi (*Aspergillus nidulans*, *Penicillium subrubescens*, *Trichoderma reesei*, *Phanerochaete chrysosporium* and *Dichomitus squalens*) using an orthology-based approach. In addition, we filtered out the potential pseudogenes and compared the expression profiles of the predicted enzyme-encoding genes based on transcriptomic, proteomic and metabolomic data of fungi grown on diverse sugars and partially validated the predicted sugar catabolic pathways with fungal growth profiles on the corresponding sugars.

## 2. Materials and Methods

### 2.1. Fungal Strains

The five fungi chosen for this study differ in their taxonomic distance to *A. niger*, ranging from very close (*A. nidulans*—a member of a different clade of the same genus) to close (*P. subrubescens*—sister genus in the order of Eurotiales) to more distant (*T. reesei*—belonging to the Sordariomycetes, a different Ascomycete class) to very distant (*P. chrysosporium* and *D. squalens*—members of a different phylum, Basidiomycota). The details of the fungal species of this study are denoted in Appendix A. 

### 2.2. Identification of Orthologs of Sugar Metabolism Genes across Six Fungal Species

Orthologs are genes in different species that originated from a common ancestor after a speciation event, resulting in the retention of similar functions during evolution. An orthology-based approach can assist us in inferring functions among different organisms. Orthologous SMGs were identified using OrthoMCL (https://github.com/stajichlab/OrthoMCL, v2.0) [15] and OrthoFinder (https://github.com/davidemms/OrthoFinder v2.5.4)) [16]. The protein sequences of the fungal species were downloaded from the JGI MycoCosm Portal (https://mycocosm.jgi.doe.gov) [17] as input for these two tools. For OrthoMCL, the protein sequences of the selected fungal genomes were compared using all-against-all BLASTP with a P-value cut-off of 1e^−5^. Many-to-many ortholog groups, including recent paralogs, were then detected based on all-against-all BLAST searches of complete proteomes as described previously [15]. The OrthoFinder method provides a fast, accurate and comprehensive platform to infer the complete set of orthologs between selected species based on the phylogenetic information from the ortho-group tree. OrthoFinder was performed using default parameters with DIAMOND [18] for sequence similarity searches and DendroBLAST [19] for the tree inference of orthogroups. Since the OrthoFinder method is an alternative and auxiliary method to OrthoMCL, it can predict orthologs that are misidentified by OrthoMCL. We merged the ortholog mapping results of these two tools as our final ortholog set. For any two ortholog groups identified by OrthoMCL and OrthoFinder that share one or more common member genes, we merged the two ortholog groups into a new group by taking the combined set of all their member genes.

### 2.3. Data Gathering and Preparation for Pathway Tools

The sugar metabolic pathway construction was carried out by the PathoLogic component of Pathway Tools v24.0 software (http://pathwaytools.com/) [20,21]. This software enables users to generate and manage an interactively edited organism-specific database (DB) named Pathway Genome Database (PGDB). The PGDB is built from an annotated genome of an organism, which integrates information about the genome, genes, proteins, biochemical reactions and predicted metabolic pathways and operons. The annotated genome of an organism was imported into PathoLogic to predict metabolic pathways. We collected annotation files derived from a wide variety of sources. The Kyoto Encyclopedia of Genes and Genomes (KEGG, https://www.kegg.jp/) annotation [22], Gene Ontology (GO) annotation [23], InterPro database annotation [24], SignalP annotation [25], Eukaryotic orthologous groups (KOG) annotation [26], Pfam [27,28] and genome functional information (.gff file) were all derived from the JGI MycoCosm Portal (https://mycocosm.jgi.doe.gov) [17]. In addition, we included the annotation of the best BLAST hit in the Swiss-Prot [29] database with an E-value <1e^−5^, as well as functional annotation obtained from our ortholog mapping analysis (Appendix A). To combine all these annotations, we created a homemade Perl pipeline to generate a set of PathoLogic format files that can be recognized by the PathoLogic component of Pathway Tools software to build databases automatically. After uploading the annotation files for PathoLogic, the program can infer metabolic pathways and enzymes by assessing the genome annotation with respect to a series of reference databases of metabolic pathways, such as EcoCyc [30] and MetaCyc databases [31]. Finally, the metabolic pathways were visualized in the corresponding organism-specific DBs, and we manually refined them by filling gaps according to ortholog mapping results and removing the predicted genes sharing low sequence homology to the reference gene of *A. niger* and excluding potential pseudogenes with extremely low expression levels (maximum FPKM < 10 in the transcriptomes of all tested conditions).

### 2.4. Transcriptome Sequencing and Analysis

The transcriptome data of *A. niger* [32] and *D. squalens* [33] were obtained from our previous studies (Gene Expression Omnibus (GEO) database accessions: GSE98572 and GSE105076). Transcriptome data of the other four species grown on nine monosaccharides were newly generated in this study. In detail, the *A. nidulans* FGSC A4 and *P. subrubescens* FBCC1632/CBS132785 were pre-cultured in complete medium [34] with 2% D-fructose, and mycelial aliquots were then transferred to minimal medium [34] with 25 mM D-glucose, D-fructose, D-galactose, D-mannose, L-rhamnose, D-xylose, L-arabinose, D-galacturonic acid or D-glucuronic acid, respectively, and cultivated for 2 h. The same cultivation approach was used for *T. reesei* QM6a and *P. chrysosporium* PR-78, but with media optimized for these species [35,36], and cultivated for 4 h. Mycelial samples were harvested after 2 or 4 h, depending on the species, and immediately frozen in liquid nitrogen.

Total RNA was extracted from ground mycelial samples using TRIzol reagent (Invitrogen, Breda, The Netherlands) according to the instructions of the manufacturer. Purification of mRNA, synthesis of cDNA library and sequencing were conducted at DOE Joint Genome Institute (JGI), as previously described [14]. The reads from each of the transcriptome sequencing (RNA-seq) samples were deposited in the Sequence Read Archive at NCBI under the accession numbers: *A. nidulans* SRP262827-SRP262853, *P. subrubescens* SRP246823-SRP246849, *T. reesei* SRP378720-SRP378745, and *P. chrysosporium* SRP249214-SRP249240. 

In this study, we filtered out the genes whose maximum expression levels (FPKM) were less than 10 in all tested conditions. 

### 2.5. Proteome Quantitation and Statistical Analysis

The sample preparation and proteomics analysis of intracellular proteins were performed similarly as previously described [37]. Briefly, the intracellular proteome was analyzed using equal amounts of proteins from each sample. MS analysis was performed using a Q-Exactive Plus mass spectrometer (Thermo Scientific, USA) outfitted with a homemade nano-electrospray ionization interface. The ion transfer tube temperature and spray voltage were 300 °C and 2.2 kV, respectively. Data were collected for 120 min following a 10 min delay after completion of sample trapping and start of gradient. FT-MS spectra were acquired from 300 to 1800 m/z at a resolution of 70 k (AGC target 3e6), and the top 12 FT-HCD-MS/MS spectra were acquired in data-dependent mode with an isolation window of 1.5 m/z at a resolution of 17.5 k (AGC target 1e5) using a normalized collision energy of 30, dynamic exclusion time of 30 s, and detected charge state of an ion 2 to 8. Generated MS/MS spectra were searched against protein sequences of each fungus obtained from MycoCosm (Appendix A) using (MSGF+) [38,39]. Best matches from the MSGF+ searches were filtered at 1% FDR, and MASIC software [40] was used to pull abundances for identified peptides. Only protein-specific peptides (peptides unique to protein in the whole protein collection) were used in consequent analysis and aggregation. InfernoR software [41] was used to transform peptide abundances (log2) and perform mean central tendency normalization. Protein-grouped normalized peptide abundances were de-logged, summed, transformed (log2) and normalized again in InfernoR to produce normalized abundances for the protein level roll-up (Appendix A). Protein abundances were then filled with zeros for missing values. The average values of nonzero abundances from the three replicates of each carbon source were used to represent the corresponding protein abundance of each growth condition. The mass spectrometry proteomics data have been deposited to the ProteomeXchange Consortium via the MassIVE partner repository with the data set identifier (MSV000090477).

### 2.6. Metabolomics Data Preprocessing and Analysis

Metabolomics analysis of intracellular metabolites was performed as previously described [42]. Briefly, metabolites were extracted from the ground mycelia underwent methoximation and silylation with N-methyl-N-trimethylsilyltrifluoroacetamide and 1% trimethylchlorosilane (MSTFA). Derivatized samples were analyzed using an Agilent GC 7890A, coupled with a single quadrupole MSD 5975C (Agilent Technologies, USA) with a standard mixture of fatty acid methyl esters (FAMEs) for RI alignment. GC-MS raw files were processed and analyzed using Metabolite Detector software, version 2.5 beta [43]. The GC/MS data have been deposited to the MassIVE database with accession number MSV000090441. A correlation analysis between the expression values of SMGs and the abundances of related metabolites was performed by using WGCNA R package [44]. Only significant correlations between SMGs and metabolites (with the absolute Pearson correlation coefficient (PCC) ≥ 0.5 and p-value < 0.05) were selected for further analysis.

### 2.7. Growth Profiling on Nine Monosaccharides

For growth phenotype analyses, strains were grown on minimal medium (MM) [34], with the exception of *T. reesei* and *P. chrysosporium,* which were grown on optimized medium for these species [35,36], on 1.5% (*w/v*) agar plates with one of nine monosaccharides (http://www.fung-growth.org/ (accessed on 1 April 2022)), including 25 mM D-glucose, D-fructose, D-galactose, D-mannose, L-rhamnose, D-xylose, L-arabinose, D-galacturonic acid and D-glucuronic acid. Growth was performed at 30 °C for both Aspergilli and 25 °C for the other species. Media with no carbon source was used as a control. If growth on a specific carbon source is the same as with no carbon source, it is considered as no growth. D-Glucose was used as an internal reference for the growth since it resulted in the best growth of all monosaccharides for the six species. Growth on other carbon sources was compared with growth on D-glucose, and the relative difference was compared between species. Growth was scored based on colony diameter and mycelium density. Growth was performed in duplicate, and no variation was observed between the duplicates for any species on any carbon source.

## 3. Results

### 3.1. Identification of Orthologs between A. niger and Five Other Fungi

We identified orthologs among three Eurotiomycota species (*A. niger*, *A. nidulans* and *P. subrubescens*), one Sordariomycota species (*T. reesei*) and two Basidiomycota species (*P. chrysosporium* and *D. squalens*) by using OrthoMCL and OrthoFinder (see Materials and Methods). Except for one gene involved in the glyoxylate cycle and one gene involved in D-gluconate metabolism, 153 sugar metabolism-related genes (SMGs) of *A. niger* had one or more copies in at least one of five tested fungi (Appendix A). Subsequently, the ortholog mapping results were incorporated into PathoLogic files to build the sugar metabolic networks.

### 3.2. Generation of Sugar Metabolic Models for the Selected Fungal Species

We established the sugar metabolic pathways for each of the selected fungi by integrating diverse annotation sources using the Pathway Tools software [20,21]. Furthermore, we excluded candidate genes with extremely low expression levels (see Materials and Methods, Appendix A). As shown in Figure 1, we summarized the total number of genes predicted to be involved in each specific sugar metabolism pathway of each studied fungus, as well as the overall completeness of the pathways based on the percentage of total reactions predicted for each pathway of each studied fungus compared to the reactions reported for *A. niger*. Overall, most sugar metabolic pathways and SMGs are highly conserved between *A. niger* and the studied fungi, while the galactose oxidoreductive pathway and D-gluconate metabolic pathways showed clear variations among the studied species, and the glycerol, L-rhamnose and PCP pathways particularly differed between the Ascomycota and Basidiomycota species (Figure 1). In the following sections, we will detail the evolutionary conservation and multi-omics profiles of the predicted sugar metabolism across different species, as well as the link between the predicted sugar metabolic network and growth profile.

### 3.3. Conservation of Sugar Metabolism among Different Fungal Species

#### 3.3.1. Sugar Catabolic Pathways with Strong Conservation

The sugar metabolic models revealed that glycolysis, TCA and glyoxylate cycles, and D-galacturonic acid catabolic pathway are highly conserved among the tested fungi (Figure 1).

Glycolysis:

Glycolysis is one of the major pathways of central metabolism and plays a key role in the growth of almost all organisms [45]. It converts glucose into pyruvate along with the formation of adenosine triphosphate (ATP) and nicotinamide adenine dinucleotide (NADH).

Typically, fungal glycolysis is a sequence of ten enzymatic reactions catalyzed by the following enzymes, namely, glucokinase (GLK, EC 2.7.1.2) [7] and hexokinase (HXK, EC 2.7.1.1) [8], glucose 6-phosphate isomerase (PGI, EC 5.3.1.9) [46], phosphofructokinase (PFK, EC 2.7.1.11) [47], D-fructose 1,6-phosphatase (FBP, EC 3.1.3.11) [48], fructose-bisphosphate aldolase (FBA, EC 4.1.2.13) [49], triphosphate isomerase (TPI, EC 5.3.1.1), glyceraldehyde 3-phosphate dehydrogenase (GPD, EC 1.2.1.12) [50], phosphoglycerate kinase (PGK, EC 2.7.2.3) [51], phosphoglycerate mutase (PGM, EC 5.4.2.11/5.4.2.12) [31], enolase (ENO, EC 4.2.1.11) [52], as well as pyruvate kinase (PKI, EC 2.7.1.40) [53]. Additionally, the metabolism of D-fructose is linked to glycolysis through hexokinase (HXK, EC 2.7.1.4), which converts D-fructose into D-fructose 6-phosphate [8,54].

Given the essential role of the glycolysis pathway, it is not surprising that all the glycolytic enzyme-encoding genes were detected in all the six studied fungi (Figure 2, Appendix A). In contrast to the completeness of glycolysis in all studied fungi, the gene copy numbers for different enzymes showed variations across different species. The PGI, PFK and PGK encoding genes are extremely conserved and present as a single copy in each fungus, while the copy numbers of other genes show a slight difference between *A. niger* and the other species. Overall, the studied Ascomycetes seem to have more similar numbers of glycolysis-related genes than the two Basidiomycetes (Appendix A). For instance, the glycolytic genes between *A. nidulans* and *A. niger* are very conserved, with almost the same numbers of orthologs identified for each enzymatic reaction step, except two more predicted genes encoding HXK/GLK and one fewer gene encoding PGM. *P. subrubescens* has more copies for both HXK/GLK and FBA. *T. reesei* has one extra copy for both HXK/GLK and PKI but one copy less for FBA, GPD and PGM. In contrast, the two Basidiomycete species only have a single copy for HXK, GLK, PGI, PFK, FBP, TPI, GPD, PGK and PKI encoding genes. However, we identified one extra copy of ENO and PGM for *P. chrysosporium*, whereas in *D. squalens*, we found extra copies of FBA and PGM. In addition, many of our predicted genes related to glycolysis in *A. nidulans* and *T. reesei* have been experimentally characterized in previous studies [55,56].

The tricarboxylic acid (TCA) and glyoxylate cycles:

Similar to glycolysis, the TCA cycle is another major pathway of central metabolism, which is essential for energy metabolism in fungi [11,12].

The input to the TCA cycle is acetyl-CoA, derived from the metabolism of carbohydrates, fatty acids and amino acids. The TCA cycle is a closed loop consisting of eight enzymatic reactions, where the oxaloacetate reacting with acetyl-CoA in the first step is reformed in the last step (Figure 2). A common source of acetyl-CoA is the oxidation of pyruvate, the end product of glycolysis. In addition, acetyl-CoA is also provided from coenzyme A and acetate through the reaction catalyzed by acetyl-CoA synthetase (AcuA, EC 6.2.1.1) [57]. Acetate used in the above reaction can be obtained from the conversion of oxaloacetate into oxalate by oxaloacetate acetylhydrolase (OAH, EC 3.7.1.1) [58]. As a modified TCA cycle, the glyoxylate cycle shares three enzymatic reactions with the TCA cycle: malate dehydrogenase (MDH, EC 1.1.1.37), aconitate hydratase (ACO, EC 4.2.1.3), and citrate synthase (CIT, EC 2.3.3.1/2.3.3.16). Therefore, the glyoxylate cycle can access the reactions of the conversion of pyruvate to oxaloacetate and acetyl-CoA, as well as the other reactions of the conversion between oxaloacetate and acetyl-CoA through the intermediate compound acetate, as described in Figure 2.

We identified almost all the necessary genes for TCA and glyoxylate cycles in the genomes of all our tested fungi, except for the OAH-encoding genes that are not present in *T. reesei* and *P. chrysosporium*. Similar to glycolysis, some of the genes are present in more than one copy (Appendix A). For example, the *acuA* gene encoding acetyl-CoA synthetase has a single copy in Basidiomycetes but has additional in-paralogs in Ascomycetes. The *acuD* gene encoding isocitrate lyase, which breaks down isocitrate into glyoxylate and succinate, has additional in-paralogs in all tested fungi except for *A. niger*. Two copies of pyruvate carboxylase (PYC, EC 6.4.1.1) were identified in the other two Eurotiomycetes, while one copy was found in *T. reesei* and the two Basidiomycetes.

D-Galacturonic acid metabolism:

D-Galacturonic acid is the main constituent of pectin and is an important carbon source for microorganisms living on plant material [59]. In fungi, D-galacturonic acid is catabolized through a non-phosphorylating pathway involved in five steps (Figure 2). After the spontaneous conversion of D-galacturonic acid to aldehyde-D-galacturonate, the latter is eventually converted into pyruvate and L-glyceraldehyde through three enzymes: D-galacturonic acid reductase (GaaA, EC 1.1.1.365), L-galactonate dehydratase (GaaB, EC 4.2.1.146), and 2-keto-3-deoxy-L-galactonate aldolase (GaaC, EC 4.1.2.54). The further conversion of L-glyceraldehyde to glycerol is catalyzed by L-glyceraldehyde reductase (GaaD/LarA, 1.1.1.372), which enters the glycerol catabolism [59,60,61,62,63,64,65].

Based on our prediction, the orthologous genes of all characterized enzymes in *A. niger* were identified in all five studied species (Appendix A). Only small variations in the number of gene copy numbers across different species were found for the D-galacturonic acid pathway. The GaaA and GaaD/LarA encoding genes are extremely conserved and are present as a single copy in each fungus. In addition, an exclusive orthologous gene encoding GaaB can be identified in the other four studied species, except for two copies in *T. reesei*. In addition, gene encoding GaaC only has a single copy in the two Eurotiomycetes but has two copies in *T. reesei* and in the two Basidiomycota species. Overall, this five-step D-galacturonic acid catabolism pathway is evolutionarily conserved in all the studied fungi (Figure 1).

#### 3.3.2. Sugar Catabolism with Moderate Conservation

In addition to the highly conserved sugar catabolic pathways described above, several sugar catabolic pathways showed moderate conservation, which include glycerol metabolism, D-mannose metabolism, pentose phosphate pathway and D-galactose catabolism. For each of these pathways, we failed to identify corresponding enzyme encoding genes for one or more essential reactions in at least one studied species.

Glycerol metabolism:

Glycerol is a ubiquitous organic compound in nature that can be metabolized by many fungi as a source of carbon and energy [66,67]. In addition, glycerol can be derived from the D-galacturonic acid catabolism pathway, as described in the section above. Before glycerol enters glycolysis, it needs to be converted into dihydroxyacetone phosphate, as shown in Figure 2.

The reversible conversion from glycerol to *sn*-glycerol 3-phosphate is mediated by glycerol kinase (GlcA, EC 2.7.1.30) and glycerol 1-phosphatase (GPP, EC 3.1.3.21). The further conversion of sn-glycerol 3-phosphate to dihydroxyacetone phosphate, which is also produced in glycolysis, is catalyzed by two different glycerol 3-phosphate dehydrogenases (GFD, EC 1.1.1.8; EC 1.1.5.3). In addition, glycerol can be converted to dihydroxyacetone and then to dihydroxyacetone phosphate by glycerol dehydrogenase (GldB, EC 1.1.1.156) and dihydroxyacetone kinase (DakA, EC 2.7.1.29), respectively [68,69,70].

Genomes of all the tested fungi encode at least one copy of each enzyme involved in this pathway, except for the two Basidiomycetes that lack *gldB*, which is required for the reduction of glycerol to dihydroxyacetone (Figure 2). The *glcA* gene had additional in-paralogs (paralogs that arose after the species split) in *A. nidulans* and *P. subrubescens*. In addition, we found multiple copies of *gldB* in *P. subrubescens*, *T. reesei* and *P. chrysosporium*, which was present as a single copy in *A. niger* and *A. nidulans*.

D-Mannose metabolism:

D-Mannose forms the backbone of polysaccharide mannan or galactomannan, which is one of the major constituents of hemicellulose in the plant cell wall [3]. Within the cell, D-mannose is phosphorylated by hexokinase (HXK, EC 2.7.1.7) to produce D-mannose 6-phosphate. The latter can either be catabolized by mannose 6-phosphate isomerase (PMI, EC 5.3.1.8) into D-fructose 6-phosphate entering into glycolysis or be converted to GDP-α-D-mannose through sequential actions of phosphomannomutase (PMM, EC 5.4.2.8) and mannose-1-phosphate guanylyltransferase (MGT, EC 2.7.7.13).

In *A. niger*, the formation of D-mannose 6-phosphate from D-mannose recruits the same *hxk* gene set involved in the conversion of D-glucose to D-glucose 6-phosphate [5,10]. However, in *T. reesei* and the two Basidiomycetes, only parts of *hxk* genes were predicted to be involved in D-mannose catabolism (Appendix A). In general, all reactions of mannose metabolism have at least one gene identified in the studied fungi, with the exception of genes involved in the conversion of α-D-mannose 1-phosphate to GDP-α-D-mannose missing in *D. squalens*.

Pentose phosphate pathway (PPP):

The pentose phosphate pathway (PPP) is an important part of central metabolism as well. It is the major source of NADPH and also provides the intermediate ribose 5-phosphate that is essential for the synthesis of nucleotides and nucleic acids [71].

The PPP is connected with D-ribulose, D-ribose and D-xylulose catabolism and produces several glycolytic intermediates, including glucose 6-phosphate, D-fructose 6-phosphate, D-glyceraldehyde-3-phosphate and NADPH that are essential components during glycolysis. The PPP can be divided into two phases, the oxidative phase and the non-oxidative phase. In the oxidative phase, glucose 6-phosphate is oxidized via three enzymatic steps to ribulose 5-phosphate with the generation of NADPH [72]. The first three reactions are catalyzed by glucose 6-phosphate dehydrogenase (GSD, EC 1.1.1.49), 6-phosphogluconolactonase (PGL, EC 3.1.1.31) and 6-phosphogluconate dehydrogenase (GND, EC 1.1.1.44). The non-oxidative branch is composed of a series of rearrangements catalyzed by an isomerase (ribose 5-phosphate isomerase, RPI, EC 5.3.1.6), an epimerase (ribulose-phosphate 3-epimerase, RPE, EC 5.1.3.1), two different transketolases (TktA, EC 2.2.1.1/TktC, EC 2.2.1.3), and a transaldolase (TAL, EC 2.2.1.2), successively involving C5 compounds (D-xylulose 5-phosphate and D-ribose 5-phosphate), C3 and C7 compounds (D-glyceraldehyde 3-phosphate and D-sedoheptulose 7-phosphate) and C4 and C6 compounds (D-erythrose 4-phosphate and D-fructofuranose 6-phosphate). In addition, the conversion between D-erythrose 4-phosphate and D-glyceraldehyde 3-phosphate can also be catalyzed by TktA. Unlike the oxidative branch, the non-oxidative branch is reversible. Another source of D-ribulose 5-phosphate and D-ribose 5-phosphate is derived from the phosphorylation of D-ribulose and D-ribose by ribulokinase (RBT, EC 2.7.1.47) and ribokinase (RBK, EC 2.7.1.15), respectively.

Except for the absence of RBK in the two studied Basidiomycota species, all reactions in the PPP have their corresponding enzymes assigned (Figure 2 and Appendix A). However, the number of genes involved in specific reactions of different fungi shows large variability (Appendix A). For example, the genes encoding GsdA and RpeA are extremely conserved. The former is present as a single copy in each fungus, and the latter has a single copy in five fungi, apart from *P. subrubescens* possessing two copies. The PglA encoding gene has a single copy in the three studied Eurotiomycetes, while additional copies were observed in other fungi. The other genes showed different patterns across different fungi. For instance, the RPI encoding gene has additional copies in the three Eurotiomycetes but a single copy in the one Sordariomycete and the two Basidiomycetes. The two TKT enzymes showed different conservation among different species. The TktC encoding gene has a single copy in five studied fungi except for *A. niger* which has one more copy. The TktA-encoding gene has a single copy in *A. niger* and *T. reesei* but two copies in the other fungi. The RbtA-encoding gene is present as two copies in *A. nidulans*, while a single copy exists in the other species.

D-Galactose metabolism:

D-Galactose is a common monosaccharide that can be utilized by all living organisms. Most microorganisms, including filamentous fungi, can utilize D-galactose for growth via three pathways: the Leloir pathway, the oxidoreductive pathway (also referred to as the De Ley–Doudoroff pathway) [73], and the non-phosphorylated De LeyDoudoroff pathway (Figure 2).

The main route of fungal D-galactose metabolism is the Leloir pathway, resulting in its conversion to glucose 6-phosphate, which can enter glycolysis (Figure 2). In the first step of this pathway, β-D-galactose, which is the most common form released during plant polysaccharides degradation, is epimerized to α-D-galactose by galactose mutarotase (also known as aldose 1-epimerase, GalM, EC 5.1.3.3). Subsequent reactions catalyzed by galactokinase (GalK, EC 2.7.1.6), D-galactose-1-phosphate uridylyl transferase (GalT, EC 2.7.7.12) and UDP-galactose 4-epimerase (GalE, EC 5.1.3.2) convert D-galactose to D-glucose 1-phosphate, which is finally transformed to D-glucose 6-phosphate by the action of phosphoglucomutase (PgmB, EC 5.4.2.2). The Leloir pathway is highly conserved, and we observed that all five enzymes involved in this pathway are present in the six studied species with at least one copy. The GalK and GalT encoding genes are extremely conserved and present as a single copy in each fungus, with the exception of two copies of *galK* in *D. squalens*. In addition, almost all genes involved in the Leloir pathway of *A. nidulans* identified in this study have been previously experimentally characterized [74].

In the oxidoreductive pathway, β-D-galactose is converted into D-fructose by a series of reductive and oxidative steps via the intermediates galactitol, L-xylo-3-hexulose, D-sorbitol and D-fructose [73]. The enzymes successively involved in this pathway include aldose reductase (EC 1.1.1.21), galactitol dehydrogenase (LadB, EC 1.1.1.-), L-xylo-3-hexulose reductase (XhrA, EC 1.1.1.-) and D-sorbitol dehydrogenase (GutB, EC 1.1.1.14). In the end, D-fructose is then phosphorylated by fructokinase (EC 2.7.1.4) to D-fructose 6-phosphate, which enters glycolysis. Previous studies in *A. niger* and *T. reesei* showed that two distinct enzymes were involved in catalyzing the first enzymatic step of the oxidoreductive pathway. In *T. reesei*, the conversion of galactitol to L-xylo-3-hexulose was mediated by L-arabinitol dehydrogenase (LAD1) [75], while in *A. niger* a different gene, the D-galactitol dehydrogenase (LadB), instead of ortholog gene of LAD1, has been identified for catalyzing this reaction [76]. Similarly, the ortholog of LadB, AN4336 in *A. nidulans* was predicted for this conversion. However, different from other species the product of galactitol oxidation was suggested as L-sorbose in *A. nidulans* [77,78,79,80]. L-sorbose was produced from D-galactitol oxidation catalyzed by LadA, and it was then reduced to D-sorbitol by the L-xylulose reductase LxrA [77,81]. In line with this hypothesis, LadA (AN0942), associated with the production of L-sorbose, was suggested in a recent study [74]. In our study, we predicted the same gene set involved in this oxidoreductive pathway in *A. nidulans*, as previously reported [79,80,82]. Although the oxidoreductive pathway for D-galactose catabolism was well established in *Aspergillus* species and *T. reesei*, knowledge of the oxidoreductive pathway in *Penicillium* and Basidiomycota species is scarce. As shown in Figure 2, the enzymes involved in the reduction of the galactitol into L-xylo-3-hexulose were not identified in *P. subrubescens* and the two Basidiomycota species. Nevertheless, we identified the potential genes related to the other four reactions, indicating the presence of a possible alternative enzymatic conversion between galactitol and L-xylo-3-hexulose in these species.

In the non-phosphorylated De Ley–Doudoroff pathway, D-galactose is eventually metabolized into pyruvate and D-glyceraldehyde-3-phosphate through five consecutive reactions producing six intermediates, including D-galactono-1,4-lactone, D-galactonate, 2-dehydro-3-deoxy-D-galactonate, pyruvate, D-glyceraldehyde and D-glyceraldehyde-3-phosphate, which are catalyzed by five specific enzymes: D-galactose dehydrogenase (EC 1.1.1.48), gamma 1,4 lactonase (EC 3.1.1.25), D-galactonate dehydratase (DGD, EC 4.2.1.6), an aldolase (EC 4.1.2.51) and a dihydroxyacetone kinase (DAK, EC 2.7.1.28). Pyruvate enters the TCA and glyoxylate cycles, while D-glyceraldehyde-3-phosphate enters glycolysis. With the exception of the enzymes synthesizing the penultimate step in this pathway, which are still elusive, the corresponding genes involved in other reactions have been identified for all species in our study. Most species harbor two D-galactose dehydrogenase encoding genes, but *T. reesei* has only one. The gamma 1,4 lactonase is highly conserved in all six tested species with one copy, while the DGD encoding gene has multiple copies, especially in *P. subrubescens* and *T. reesei,* which contain seven and eight copies, respectively. The DAK encoding gene was detected in all species with one copy except in *T. reesei,* with two duplicates.

Overall, the D-galactose metabolism is relatively conserved in all studied species, despite several specific enzymes encoding genes that differ in copy numbers depending on the species and their taxonomic distances. Notably, the predicted gene content of the D-galactose metabolism pathway in the two Basidiomycota species showed the most differences from the four Ascomycota species (Appendix A).

#### 3.3.3. Sugar Catabolism with Low Conservation

Pentose catabolic pathway (PCP):

L-Arabinose and D-xylose are the most abundant pentoses found in the hemicelluloses (such as (arabino)xylan and xyloglucan) and pectin of the plant cell wall [77]. Typically, L-arabinose and D-xylose are metabolized by the fungus through the pentose catabolic pathway (PCP), consisting of oxidation, reduction and phosphorylation reactions to form D-xylulose-5-phosphate, which enters the PPP.

PCP is present in the majority of filamentous fungi, although there are some differences in the enzymes that catabolize each specific step. Traditionally, only a single enzyme was assigned to catalyze each reaction in PCP. A recent study revealed the redundancy and complexity of the conversion of pentose sugars in *A. niger* [9]. L-arabinose goes through the sequential reactions: reduction catalyzed by NADPH-dependent L-arabinose reductase (LarA) and D-xylose reductase (XyrA/XyrB) (EC 1.1.1.21); oxidation catalyzed by L-arabitol-4-dehydrogenase (LadA, EC 1.1.1.12), xylitol dehydrogenase (XdhA) and sorbitol dehydrogenase (SdhA) (EC 1.1.1.12); reduction catalyzed by L-xylulose reductases (LxrA/LxrB, EC 1.1.1.10) [83] and again oxidation (LadA/XdhA/SdhA, EC 1.1.1.9), by which L-arabinose is progressively converted to L-arabitol, L-xylulose, xylitol and D-xylulose (Figure 2). D-xylulose is further phosphorylated into D-xylulose 5-phosphate by D-xylulose kinase (XkiA, EC 2.7.1.17).

D-Xylose metabolism starts with the reduction of D-xylose into xylitol catalyzed by the same enzymes (LarA/XyrA/XyrB, EC 1.1.1.307) involved in the reduction of L-arabinose. Xylitol is further converted into D-xylulose and D-xylulose 5-phosphate sequentially, as described in L-arabinose metabolism. Based on our prediction, the genes encoding the enzymes for the D-xylose catabolic pathways are present in all studied species, in some cases as multiple copies, particularly the D-xylose reductase and xylitol dehydrogenase genes (Appendix A). In contrast, not all the genes involved in the metabolism of L-arabinose are conserved across our studied fungi, e.g., no orthologs of *A. niger* LadA and LxrA/LxrB genes were detected in the two Basidiomycota species. In addition, instead of two copies of the gene encoding LadA in *A. niger*, only a single LadA gene was found in the other two studied Eurotiomycetes (*A. nidulans* and *P. subrubescens*) and in one Sordariomycete species (*T. reesei*).

Given the resemblance between the reduction of L-arabinose and the first reduction reaction of the D-galactose oxidoreductive pathway, it is not a surprise that the same enzymes were reported for catalyzing the production of L-arabitol/xylitol and galactitol in *T. reesei* and *A. nidulans* [76,77,84]. In this study, we predicted similar functions of these enzyme-encoding genes for the other four studied species (Appendix A).

Intriguingly, we predicted that the orthologs of *A. niger* LadA and LadB seem to have broader functions in other studied species than their predicted or characterized functions in *A. niger*. For example, we predicted that LAD1 (TrB1462W) of *T. reesei*, homologous to *A. niger* LadA, is involved in four enzymatic conversions: L-arabitol to L-xylulose, xylitol to D-xylulose, galactitol to L-xylo-3-hexulose and D-sorbitol to keto-D-fructose. The two reactions, L-arabitol to L-xylulose and galactitol to L-xylo-3-hexulose, were previously reported [84]. Exceptions are the conversions of galactitol to L-xylo-3-hexulose in *P. subrubescens* and the two Basidiomycota species, in which other genes seem to take over this function (Appendix A). Similarly, LadB (AN4336) in *A. nidulans* was predicted to be involved in multiple reactions (Appendix A).

L-Rhamnose metabolism:

L-Rhamnose is enriched in some fractions of plant biomass, such as hemicellulose and pectin. In fungi, L-rhamnose is generally degraded in four consecutive reactions catalyzed by an NADH-dependent L-rhamnose-1-dehydrogenase (LraA, EC 1.1.1.173), L-rhamnonic acid lactonase (LrlA, EC 3.1.1.65), L-rhamnonate dehydratase (LrdA, EC 4.2.1.90) and 2-keto-3-deoxy-L-rhamnonate aldolase (LkaA, EC 4.1.2.53), resulting in the formation of L-rhamnose-1,4-lactone, L-rhamnonate, 2-dehydro-3-deoxy-l-rhamnonate, pyruvate and L-lactaldehyde [14,85].

All genes involved in the four enzymatic steps of this pathway have been first identified in the yeast *Scheffersomyces (Pichia) stipitis* [86], and the *A. niger* L-rhamnose catabolic genes have also recently been characterized [14,85]. Therefore, based on six known genes in *A. niger* (Appendix A), we predicted the genes associated with this pathway in the other species. The genes encoding enzymes involved in the first and last (fourth) reactions were highly conserved in all six studied species, except the two copies of the genes encoding LraA and LkaA observed for *A. niger* and a single gene for other species. However, the LrlA and LrdA encoding genes involved in the second and third reactions were only conserved in three studied Ascomycetes but missing in the two studied Basidiomycetes (Appendix A), which may indicate the presence of possible alternative enzymes involved in these reactions in the Basidiomycetes. In addition, we identified the same enzyme candidates of L-rhamnose metabolism in *A. nidulans* as in a previous study [87], with the exception of LkaA.

D-Gluconate metabolism:

A wide group of filamentous fungi has the ability to produce gluconic acids, such as *Aspergillus* and *Penicillium* [88]. In *A. niger*, D-gluconate catabolism includes five enzymatic reactions which are catalyzed by five enzymes: glucose oxidase (GOX, EC 1.1.3.4), D-glucono-1,5-lactone lactonodehydrolase or gluconolactonase (EC 3.1.1.17), gluconate dehydratase (EC 4.2.1.140), 2-dehydro-3-deoxy-D-gluconate D-glyceraldehyde-lyase (EC 4.1.2.51) and gluconate kinase (GukA, EC 2.7.1.12).

In all the studied species, missing enzymes in the D-gluconate catabolism were observed (Figure 2). Furthermore, the gene(s) involved in the conversion of 2-keto-3-deoxy-D-gluconate to pyruvate remains unknown among all the studied fungi. Apart from *P. subrubescens*, the other fungi have one or more copies of glucose oxidase (GOX). The highest number of GOX encoding genes was identified in *P. chrysosporium* with five genes, followed by *D. squalens* with three genes. The genes encoding gluconolactonase (EC 3.1.1.17) involved in the second step were present with more than two copies in most species except *P. chrysosporium*. The gluconate dehydratase (EC 4.2.1.140) is highly conserved in all species. In addition, D-gluconate can enter the PPP through catalysis by GukA, which was identified in most of the studied fungi except *D. squalens*.

### 3.4. The Expression Profiling of Sugar Metabolic Genes in Each Fungus

In order to explore the transcriptional responses of different fungi to distinct monosaccharides, we performed RNA-seq experiments from fungal cultivations on nine monosaccharides. The clustering results indicate the similarity of the overall expression profile across different growth conditions. The hexose (D-glucose, D-mannose and D-fructose) and pentose (D-xylose and L-arabinose) conditions were well clustered to each other, respectively, on the transcriptome data of the three Eurotiomycetes (*A. niger*, *A. nidulans*, *P. subrubescens*) (Figure 3 and Appendix A), while in the other species L-arabinose were clustered more closely with D-galactose than with D-xylose. In addition, the samples of D-galacturonic acid and D-glucuronic acid were also closely clustered together for all studied species.

Furthermore, we observed obvious sugar-specific inducing patterns for many SMGs during fungal growth on the corresponding sugars (Figure 3 and Appendix A). These clear inducing patterns of specific sugar metabolic genes on corresponding growth conditions support their predicted function in our sugar metabolic models. For example, the predicted galacturonic acid metabolism-related genes were significantly more highly expressed during fungal growth on galacturonic acid than their expression on other sugar conditions for all tested fungi, except for *D. squalens* which lacks the corresponding transcriptome data. In addition, many of the L-rhamnose and PCP-related genes were induced on the corresponding growth conditions in *A. niger*, *A. nidulans*, *P. subrubescens* and *T. reesei*. A few D-galactose oxidoreductive pathway-related genes were also induced for the three tested Eurotiomycetes, while no clear inducing patterns were observed for the other tested species. Several genes that were involved in both PCP and the D-galactose oxidoreductive pathway were highly expressed in D-xylose, L-arabinose and D-galactose in *A. nidualans*, *P. subrubescens* and *T. reesei*. In contrast to the evident sugar-specific inducing patterns of SMGs observed for Ascomycetes, only a limited number of SMGs induced by a corresponding sugar were observed for the two studied Basidiomycetes.

### 3.5. Proteome Profile of Sugar Metabolism-Related Genes (SMGs) Showed Similar Sugar Inducing Pattern as Transcriptome Data

The intracellular proteome of the studied fungi (except *D. squalens*) grown on each monosaccharide condition was also analyzed to reveal the fungal response to each specific sugar at the proteome level. In general, the protein abundance profile of SMGs showed higher similarity among replicates of the same sugar, as well as more stable production among different sugars compared to the corresponding production profiles analyzed using the whole proteome (Appendix A).

The comparison of the protein abundance profile of each specific sugar pathway among different sugars showed similar patterns as observed in the transcriptome data. The abundance profiles of proteins involved in sugar pathways of the D-galacturonic acid, L-rhamnose and PCP in the studied Ascomycetes (*A. niger*, *A. nidulans*, *P. subrubescens* and *T. reesei*) match their predicted functions. As observed, the protein abundances of SMGs involved in the above pathways were higher in corresponding sugars than in other tested sugars (Figure 4A and Appendix A). In contrast, only proteins involved in the D-galacturonic acid pathway showed strong sugar specificity in *P. chrysosporium*, while the sugar-inducing patterns of other SMGs were not obvious in the proteome data (Figure 4B). For other studied sugar pathways (e.g., TCA, glycolysis and PPP), we did not observe a clear difference in protein abundance among sugars.

### 3.6. Correlation between the Abundance of Metabolites and Gene Expression Levels of Sugar Metabolism-Related Genes (SMGs)

To further verify the accuracy of our predicted sugar metabolic pathways and investigate the association between metabolites and SMGs, we compared the metabolomics profiles across different sugar conditions and performed the correlation analysis between the abundance of SMGs and metabolites in each fungus (except *D. squalens* that lacks related data). Overall, we detected the sugars taken up by the cell, sugar metabolic intermediates and several storage compounds for each fungus (Appendix A). Several intermediates were consistently detected with higher abundance in corresponding sugar conditions for most of the studied fungi, such as xylitol, galactitol and galactonic acid showed higher abundance in D-xylose, D-galactose and D-galacturonic acid conditions for most of the studied fungi. The commonly detected storage compounds include erythritol, mannitol and ribitol.

Consistent with stronger inducing patterns of SMGs identified in the transcriptome and proteome profiles of Ascomycetes compared to Basidiomycete, we detected that more SMGs showed a stronger correlation with the corresponding metabolites in four studied Ascomycetes than in *P. chrysosporium* (Figure 5 and Appendix A). For instance, in total, 21 SMGs in *A. nidulans* (mainly including PCP, D-galactose, D-galacturonic acid, L-rhamnose pathways and glycolysis) showed strong positive correlations (correlation coefficient ≥ 0.8) with their corresponding metabolites (Figure 5). However, only two SMGs displayed high positive correlations with the metabolites in *P. chrysosporium*. In the other two studied Eurotiomycetes species, we identified 14, 1, 2, 1 and 1 SMGs, respectively, involved in the PCP, D-galacturonic acid, L-rhamnose, glycolysis and glycerol pathways in *A. niger*, as well as 3, 3, 2 and 4 SMGs, respectively, involved in the PCP, D-galacturonic acid, L-rhamnose and D-galactose pathways in *P. subrubescens*, which strongly correlated to detection of the related metabolites (Appendix A). As for *T. reesei*, we discovered six PCP-related SMGs, which have high correlations with xylitol (1), D-arabinose (3) and D-xylose (2), two glycolysis-related SMGs highly associated with D-glucose (1) and D-glucose-6-phosphate (1), but no SMGs showed high correlations with rhamnose (Appendix A). Taken together, the abundance profile of SMGs and metabolites confirmed the predicted sugar metabolic pathways and the diversity of fungal responses to different sugars.

### 3.7. Comparative Growth Profiles on Different Carbon Sources

We performed growth profiling of the six fungi on ten different monosaccharides as solo carbon sources to analyze the influence of different sugars on their growth. Given that the completeness and efficiency of sugar metabolic pathways are directly linked with their growth on the corresponding sugar, we expect clear differences in growth profiles to be observed when two different fungi with predicted different completeness of specific sugar metabolic pathways are grown on the same sugar. Therefore, the comparison of the growth profiles of these fungi on different sugars could partially confirm the accuracy of our predicted sugar metabolic pathways.

The growth profiles of the six fungi showed both notable similarities and differences in different sugars (Figure 6). In line with the complete sugar metabolic pathways of all tested sugars predicted for *A. niger* and the other two studied Eurotiomycetes species (*A. nidulans* and *P. subrubescens*), they showed similar growth profiles on all tested carbon sources. In addition, we noticed good growth for all six species on D-glucose, D-fructose and D-mannose compared to their growth on a no-carbon source, which matches the conservation and completeness of the sugar metabolic pathways of these hexoses across all studied species. In contrast, different species show varying degrees of growth on L-rhamnose. *A. nidulans* and *P. subrubescens* grew better on this monosaccharide than *T. reesei*, while the two Basidiomycota species (*P. chrysosporium* and *D. squalens*) grew poorly on L-rhamnose, which can be explained by the missing enzymes of L-rhamnose metabolic pathway predicted for these species in our study. In addition, there is a clear difference in the growth of L-arabinose between the Ascomycota and Basidiomycota species. The poor growth of L-arabinose for *P. chrysosporium* and *D. squalens* could be linked to the predicted absence of both *ladA* and *lxrA/lxrB* encoding genes in their genomes. Although orthologues of *xdhA* and *sdhA* were detected in the two Basidiomycete species, they cannot fully compensate for the loss of *ladA*, which seems to play a key role in L-arabinose metabolism. In particular, growth on L-arabinose was almost abolished for *P. chrysosporium*.

In addition to the consistency observed between the growth profile and predicted sugar pathway models, we also observed the scenario that despite the fact that fungi possess a full enzyme set in a sugar pathway, it does not ensure their good growth on the related sugar. For example, although the genes related to D-xylose metabolism were present in all fungi, the two Basidiomycota species showed poor growth on D-xylose. A similar profile was observed for the poor growth of *T. reesei* on D-galacturonic acid, although the full enzyme set to metabolize the D-galacturonic acid is present in its genome. In *A. niger* and *A. nidulans*, all genes involved in D-ribose metabolism were predicted, but their growth on D-ribose was poor. However, *P. subrubescens* and *T. reesei* grew well on D-ribose, which is in line with the presence of the related genes in their genome. Due to the absence of ribokinase encoding genes in *P. chrysosporium* and *D. squalens*, poor growth on D-ribose was observed. A great difference in growth on D-galactose was observed between *A. niger*, the other two Eurotiomycetes species and the two Basidiomycota species. Although *A. niger* and *T. reesei* have a full set of D-galactose metabolic enzymes, the growth of *A. niger* and *T. reesei* on D-galactose was poor. *P. chrysosporium* and *D. squalens* also grew poorly on D-galactose, while *A. nidulans* and *P. subrubescens* displayed relatively good growth on D-galactose.

## 4. Discussion

We successfully established sugar metabolic networks among six diverse fungi using an orthology-based approach and integrating a broad range of functional annotation and transcriptome data. The difference in predicted sugar metabolic pathways between the reference species (*A. niger*) and the other five studied fungi is well correlated with their taxonomic distances, e.g., most sugar catabolic pathways were highly conserved between the other two studied Eurotiomycetes and *A. niger*, while a larger level of diversity was observed between the Sordariomycete species and *A. niger*, and even more so for the Basidiomycete species. The predicted sugar metabolic pathways and their corresponding genes were partially supported by the growth profiles and multi-omics profiles of fungi grown on diverse sugars.

Many of our predicted SMGs in the two well-studied model fungi (*A. nidulans* and *T. reesei*) have already been experimentally characterized (Appendix A), which confirmed the accuracy of our prediction. For the less studied *P. subrubescens* and the two Basidiomycete species, we provided the first sugar metabolic blueprint of these species.

Compared to the fungal sugar metabolic pathway information available in the well-known The Kyoto Encyclopedia of Genes and Genomes (KEGG) database (https://www.kegg.jp/ (accessed on 1 May 2022)), we provided a more comprehensive and detailed sugar metabolic network. For instance, there is no specific pathway assigned to D-galacturonic acid metabolism in the KEGG database, and the galactose metabolic pathway (KEGG pathway: map00052) was not explicitly delineated into three different sub-pathways. Some crucial reactions of PCP and PPP pathways (KEGG pathway: map00030 and map00040, respectively) were not well-annotated. For example, four reactions (EC, 2.7.1.47/1.1.3.4/4.2.1.140/4.1.2.51) involved in PPP and three reactions (EC, 1.1.1.21/1.1.1.12/1.1.1.10) in PCP were absent in KEGG for *A. nidulans* and *T. reesei*.

Several sugar metabolic pathways were highly conserved across all studied fungi, e.g., TCA, glycolysis, PPP and D-galacturonic acid metabolism. Other pathways showed large variations across the different fungi, including L-rhamnose and L-arabinose catabolism. However, the completeness of the predicted pathways cannot ensure a very similar transcriptome response and growth profile of the fungi grown on corresponding sugars. For instance, although the completeness of pathways was identified for D-fructose and D-galacturonic acid in all studied fungi, we observed the relatively poor growth of *P. chrysosporium* on D-fructose, poor growth of *T. reesei* and *P. chrysosporium* on D-galacturonic acid. This could be related to the difference in catalytic efficiency and regulation mechanisms of corresponding metabolic enzymes in different species. In addition, the efficiency of fungal sugar utilization is not only related to metabolic enzymes but also to the sugar transport system [89]. One of the best examples to support this hypothesis is that the poor growth of *A. niger* on D-galactose was mainly due to the inactivity of relevant transport and germination triggers [90,91,92]. On the other hand, the predicted incompleteness of a metabolic pathway for sugar does not always mean a poor growth profile. For example, we failed to identify one key enzyme converting L-xylulose to xylitol in L-arabinose catabolism in the two Basidiomycota species; however, the corresponding intermediates (e.g., arabitol, xylitol and D-xylulose) were detected in metabolomics analysis of *P. chrysosporium* and the growth profile showed that they can utilize L-arabinose. This indicates that possible new enzymes may be involved in this pathway. The failure of the identification of these alternative enzymes in our study could be due to their low sequence similarity to the enzymes predicted or validated in *A. niger*. An alternative approach integrating sequence features and multiple omics data using machine learning could be used to detect those enzymes [93,94,95].

The different transcriptome and proteome profiles of SMGs between Ascomycota and Basidiomycota species could be related to their different transcriptional regulation and be linked with their adaption to different ecological niches. In line with this, previous studies have demonstrated a dramatic difference in the repertoire of regulatory proteins of Ascomycete and Basidiomycete fungi [96,97]. Orthologs of the well-studied transcription factors controlling specific SMGs in Ascomycete fungi, e.g., XlnR, AraR, RhaR, GaaR, and GalX, were missing in the Basidiomycete fungi. The lack of clear sugar-inducing patterns in the Basidiomycota species could also suggest that in their natural environment, the studied Basidiomycetes rarely experience the relatively high monomeric-sugar concentrations that were used in our study.

## Figures and Tables

**Figure 1 jof-08-01315-f001:**
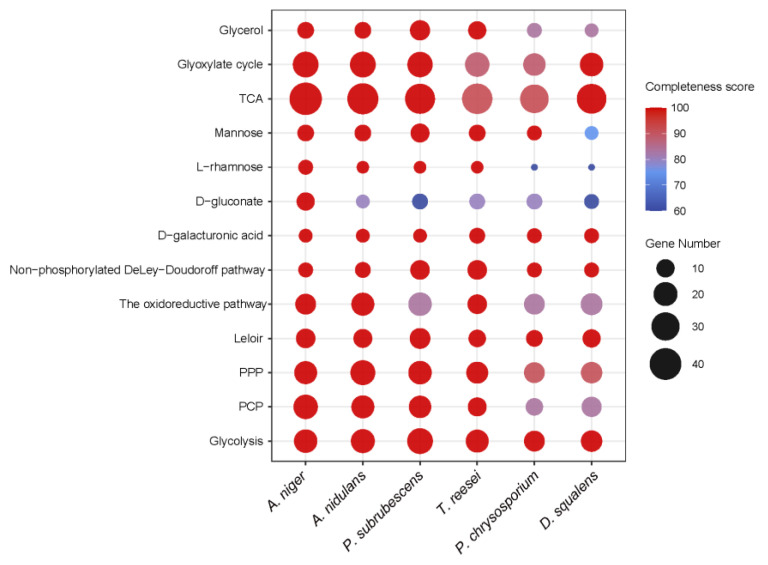
Conservation of sugar metabolic pathways among *A. niger*, *A. nidulans*, *T. reesei*, *P. subrubescens*, *P. chrysosporium* and *D. squalens*. The size of the dots indicates the number of genes involved in each pathway, and the color indicates the completeness of the pathway. The completeness score of a pathway is defined as the percentage of predicted reactions in a studied fungus compared to the total reactions reported in *A. niger* for each specific sugar metabolic pathway.

**Figure 2 jof-08-01315-f002:**
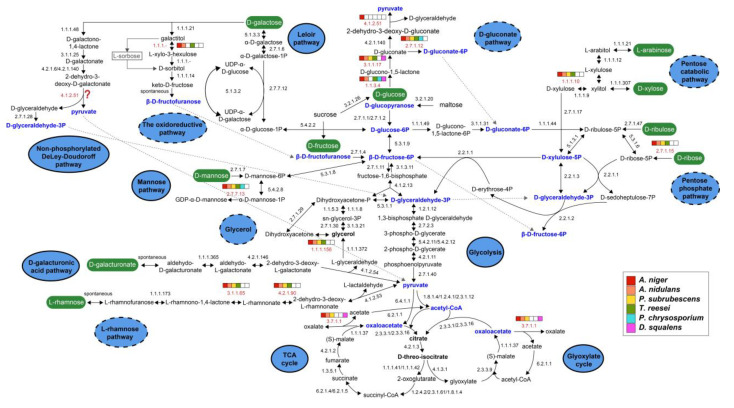
Sugar metabolic networks of *A. niger*, *A. nidulans*, *T. reesei*, *P. subrubescens*, *P. chrysosporium* and *D. squalens*. The names of the sugar metabolic pathways are shown in blue circles. The solid or dashed lines of the circles indicate whether the corresponding pathway is complete or not, respectively. For enzymes involved in each reaction of each pathway, the corresponding enzyme EC number is shown, and for the reaction with the absence of genes in one or more of our studied species, the EC number is highlighted in red. A color bar was also used to highlight the presence and absence of genes in specific reactions in the studied species. The question marks indicate that the corresponding reaction remains to be discovered.

**Figure 3 jof-08-01315-f003:**
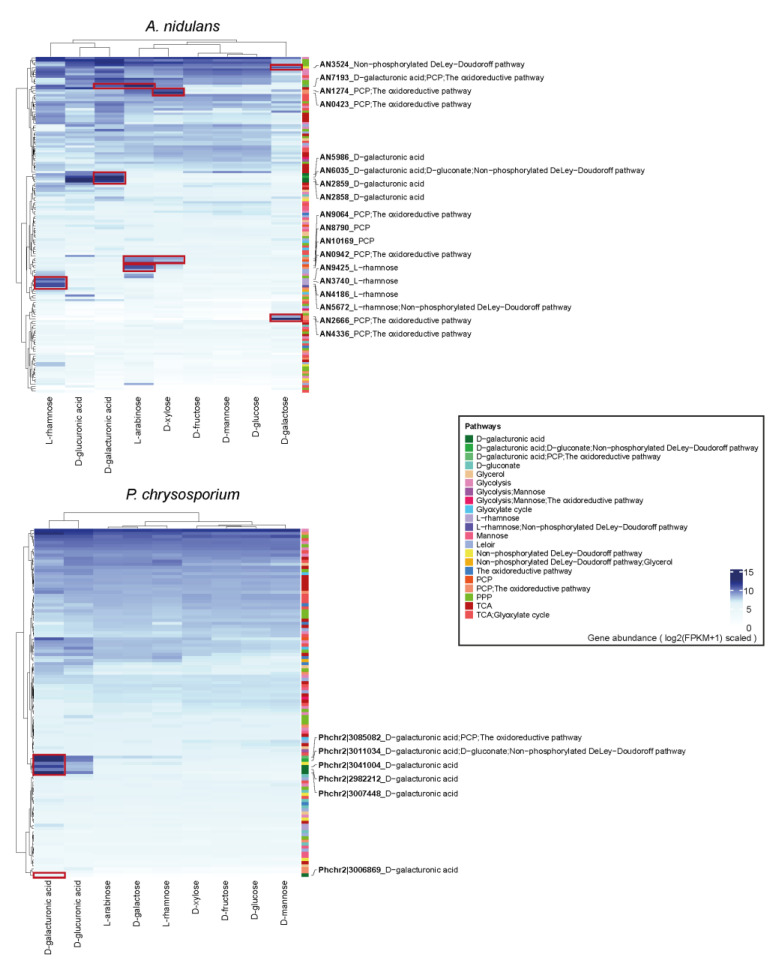
Expression profiles of sugar metabolism-related genes in *A. nidulans* and *P. chrysosporium* during their growth on diverse monosaccharides. The blue color from light to dark indicates a gene expression level from low to high. Selected genes with specific sugar-induced expression patterns are marked with a red box on the heatmap, and their gene IDs and the associated pathways are displayed. On the right bar, different colors indicate different pathways.

**Figure 4 jof-08-01315-f004:**
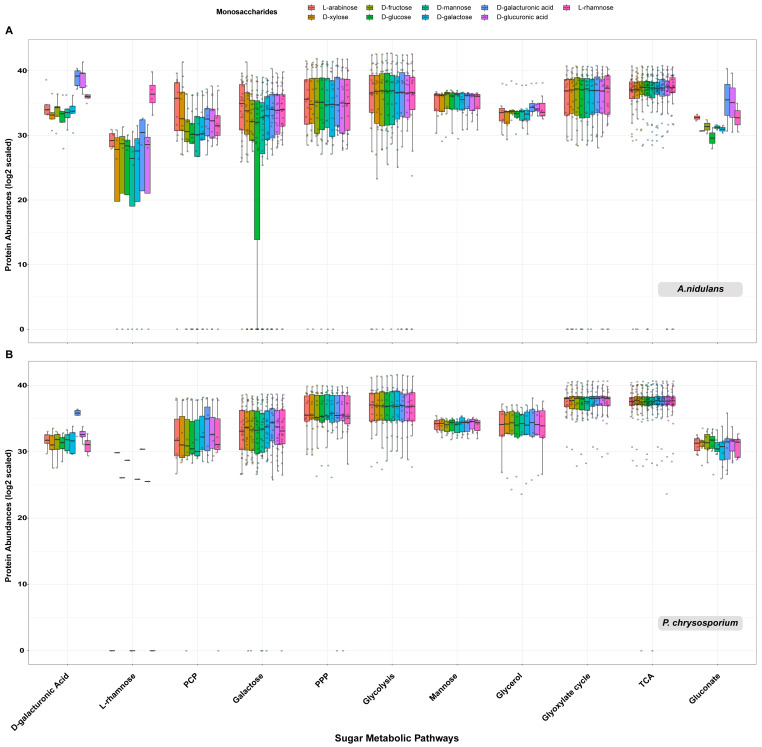
Protein abundance profiles of the sugar metabolism-related proteins involved in each sugar metabolic pathway in *A. nidulans* (**A**) and *P. chrysosporium* (**B**). On the boxplot, different colors indicate different monosaccharide growth conditions, and each small circle indicates an individual protein related to each specific sugar metabolic pathway. The metabolic enzymes that were detected in less than one-third of the tested growth conditions were not included in the boxplots. The y-axis represents the abundance of proteins (log2 scaled), and the x-axis depicts different sugar metabolic pathways.

**Figure 5 jof-08-01315-f005:**
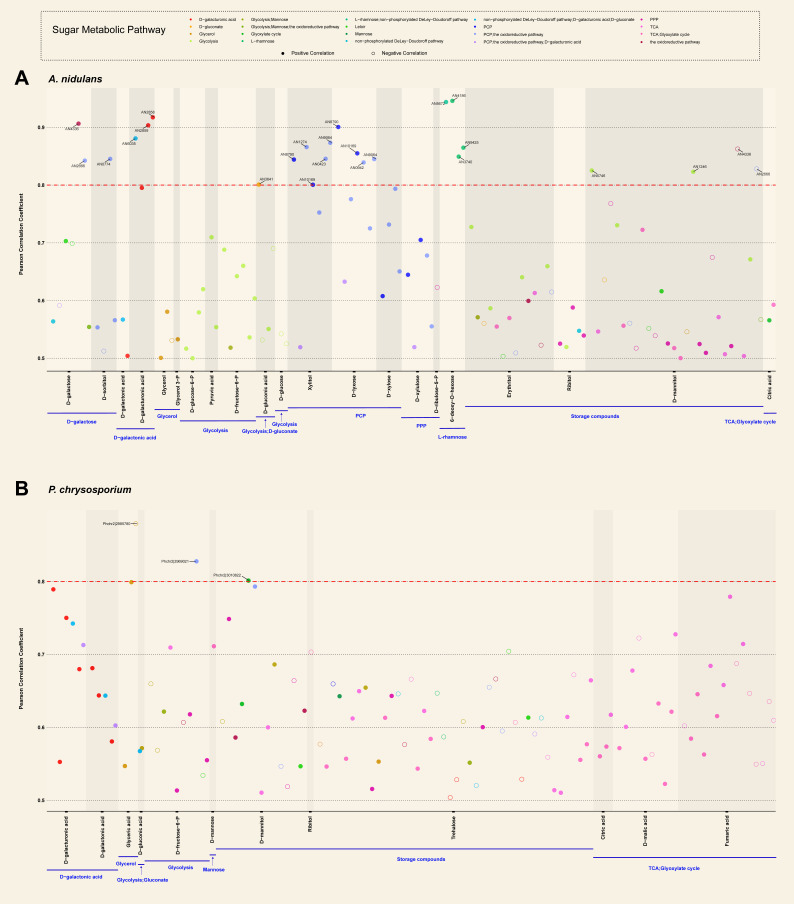
Correlation between the abundance of metabolites and sugar metabolism-related genes in *A. nidulans* (**A**) and *P. chrysosporium* (**B**). The y-axis represents Pearson correlation coefficient (PCC ≥ 0.5), and x-axis depicts different metabolites and the related sugar metabolic pathways (highlighted in blue). Each dot represents a gene, and its color indicates the corresponding sugar pathway in which it is involved. The positive and negative correlations were shown in filled and open circles, respectively. Only the names of genes with high correlation (PCC ≥ 0.8) to the analyzed metabolites are displayed.

**Figure 6 jof-08-01315-f006:**
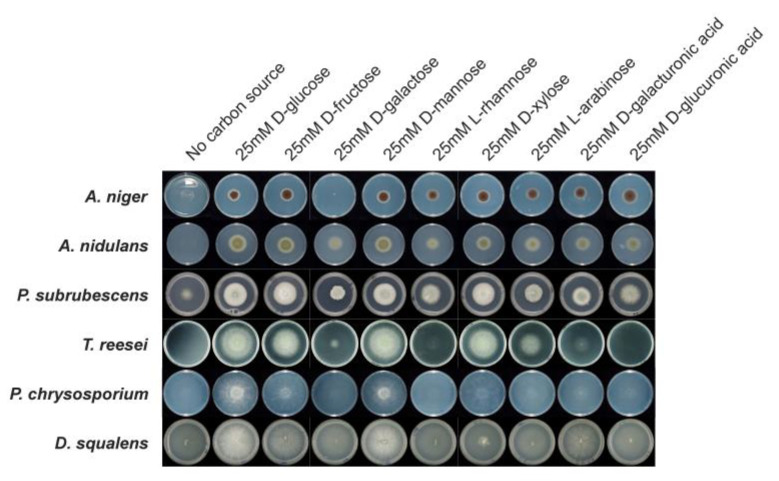
Fungal growth profiling on different monosaccharides. All strains were grown on minimal medium (MM) with nine different carbon sources. Growth was performed at 30 °C for the two *Aspergilli* and 25 °C for other species. If growth on a specific carbon source is the same as with no carbon source, it is considered no growth.

## Data Availability

The reads from each of the transcriptome sequencing (RNA-seq) samples were deposited in the Sequence Read Archive at NCBI under the accession numbers: *A. nidulans* SRP262827-SRP262853, *P. subrubescens* SRP246823-SRP246849, *T. reesei* SRP378720-SRP378745, and *P. chrysosporium* SRP249214-SRP249240. The mass spectrometry proteomics data have been deposited to the ProteomeXchange Consortium via the MassIVE partner repository with the data set identifier (MSV000090477). The GCMS metabolomics data have been deposited to the MassIVE database with accession number MSV000090441.

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
