# Peer review of "The Sugar Metabolic Model of *Aspergillus niger* Can Only Be Reliably Transferred to Fungi of Its Phylum"

_jof, 2022, doi:10.3390/jof8121315_

Round 1

Reviewer 1 Report

Dear Authors,

congratulations for your great work...

Just few suggestions I can give you for slight modification of the work...

Very interesting and very hard work can be seen from the MS 

see comment in the pdf

Reviewer 2 Report

Manuscript entitled: The Sugar Metabolic Model of Aspergillus niger Can Only be

Reliable Transferred to Fungi of Its Phylum. This manuscript mapped the well-established sugar metabolic network of Aspergillus niger to five taxonomically distant speciesusing an orthology-based approach. This manuscript had significance on facilitating rational metabolic engineering of these fungi as microbial cell factories. Minor points need to be addressed before published.

1. Line 98. “P-value cut-off of 1e-5”. -5 needs superscript.

2. Section 2.7. Line 198-206

How the author compared the growth ability of these strains under different carbon sources? Using colony diameter? How many replications were conducted? Are there any statistic analysis for the growth ability of strains?

3. Line 209-211. These contents need remove to section 2.2.
